# Controversies and Clarifications Regarding the Role of Aspirin in Preeclampsia Prevention: A Focused Review

**DOI:** 10.3390/jcm13154427

**Published:** 2024-07-29

**Authors:** Amihai Rottenstreich

**Affiliations:** 1Laboratory of Blood and Vascular Biology, Rockefeller University, New York, NY 10065, USA; amichaimd@gmail.com; Tel.: +1-212-327-7494; Fax: +1-212-327-7493; 2Division of Maternal-Fetal Medicine, Department of Obstetrics and Gynecology, Zucker School of Medicine at Hofstra/Northwell, New York, NY, USA

**Keywords:** preeclampsia, aspirin, prevention

## Abstract

Preeclampsia is one of the leading causes of maternal and perinatal morbidity and mortality worldwide. In recent decades, many studies have evaluated different interventions in order to prevent the occurrence of preeclampsia. Among these, administration of low-dose aspirin from early pregnancy showed consistent evidence of its prophylactic role. In this article, we review the scientific literature on this topic, highlighting the rationale for aspirin use, who should be treated, the timing of initiation and cessation of therapy, the importance of proper dosing, and its role in the prevention of other adverse outcomes.

## 1. Introduction

Preeclampsia, affecting 2–8% of all pregnancies, is one of the leading global causes of maternal and perinatal morbidity and mortality [1,2]. Preeclampsia accounts for one-sixth of all preterm births and is frequently associated with fetal growth restriction, thus potentially leading to lifelong consequences for the offspring [3,4]. In addition, mothers affected by preeclampsia have significantly increased risks of cardiovascular and cerebrovascular disease later in life [5,6,7]. In recent decades, many studies have focused on elucidating the pathophysiology underlying preeclampsia, identifying those at risk, and evaluating prophylactic interventions to prevent its occurrence. In this article, we review the scientific literature regarding low-dose aspirin (LDA) use in preeclampsia prevention, highlighting the rationale for the use of aspirin, who should be treated, the timing of initiation and cessation of therapy, and the importance of proper dosing. A summary of this review is given in Table 1.

## 2. History of Aspirin

Aspirin-related compounds extracted from the willow tree have been used for over 3000 years [8]. Reports of its use can be found in Egyptian papyrus scrolls dating back to 1534 BCE [9]. The Greek physicians Galen and Hippocrates described in 400 BC that the willow extract could be used to relieve headache, pain and fever [10]. However, salicylic acid was first synthesized in a laboratory in 1853; in 1897 it was modified to form acetylsalicylic acid and named aspirin. First industrial production of aspirin started in 1915, with the 1918 flu pandemic leading to widespread use of aspirin [11]. In 1982, the Nobel Prize was awarded to Vane, Samuelsson, and Bergstrom for elucidating the mechanism of action of aspirin [12]. Aspirin was shown to exert its analgesic, antipyretic, and anti-inflammatory effects through inactivation of the enzymes cyclooxygenase (COX)-1 and COX-2, thereby suppressing the production of prostaglandins and thromboxane A2 production. The reduction in the latter also accounts for aspirin’s antithrombotic effects by inhibiting arachidonic acid-induced platelet aggregation [13]. Along with its antiplatelet and anti-inflammatory effects, the postulated mechanisms through which aspirin can prevent preeclampsia include direct positive effects on the placentation process, allowing proper remodeling of the spiral arteries, and this is supported by in-vitro data showing the beneficial effects of aspirin on trophoblast function, reduction of apoptosis, and improvement in placental angiogenic factors [14].

## 3. Aspirin in the Prevention of Preeclampsia

The first report to suggest a possible positive effect of aspirin on preeclampsia was published in 1978, in which daily use of aspirin in a woman with two prior pregnancies affected by preeclampsia and fetal growth gestation led to improved pregnancy outcomes [15]. While the dose and timing of aspirin initiation in this report differ significantly from current clinical practice, it sparked interest in the potential role of aspirin in preeclampsia prevention. Starting in 1985 [16], a series of randomized controlled trials (RCTs) was conducted to evaluate the role of aspirin in the prevention of placenta-mediated complications, including preeclampsia, with inconsistent results reported. The PARIS individualized participant data meta-analysis published in 2007 included 24 RCTs comparing aspirin to no treatment or placebo and reported a 10% risk reduction (relative risk [RR] 0.90, 95% confidence interval [CI] 0.84–0.97) in the incidence of preeclampsia [17]. The relatively modest effect found in the PARIS meta-analysis can be attributed to the heterogeneity in the design of the included studies: studies evaluating aspirin at any given dose (range 50–150 mg, median 75 mg) or initiated at any time throughout gestation (mostly after 20 weeks). There was also a lack of standardization of the disease definition with 15 different definitions among the studies included [17]. Several subsequent meta-analyses have demonstrated that LDA is highly effective in the prevention of *preterm* preeclampsia (occurring prior to 37 weeks) only if therapy is initiated before 16 weeks gestation, with no conferred benefit on *term* preeclampsia [18,19,20,21].

### 3.1. Who Should Be Treated with LDA in Order to Prevent Preeclampsia?

Considering the positive effect of aspirin, a key issue is how to identify those at risk of developing preeclampsia. Given that the beneficial effect of aspirin is maximized when initiated before 16 weeks gestation, it is critical to perform such screening no later than the first trimester. Various methods have been proposed to identify the target population. One approach is maternal-characteristics-based screening, as recommended by the American College of Obstetricians and Gynecologists (ACOG) and the Society of Maternal-Fetal Medicine (SMFM) [22]. This risk stratification is based only on risk factors that can be obtained from a patient’s medical history, with aspirin recommended in those with at least one high-risk factor or more than one moderate-risk factor [22,23,24]. The main advantage of this approach is its simplicity. However, it was shown to identify only 41% of cases of preterm preeclampsia, with a 64% screen-positive rate [25]. Moreover, the use of LDA solely based on the presence of the different medical conditions included in the recommendations of the ACOG and SMFM is not supported by level 1 data. For example, the use of aspirin in those with pre-gestational diabetes and chronic hypertension, each of which represents a high-risk factor in these recommendations did not result in lower rates of preeclampsia in several studies [26,27,28,29].

Innovatively, the Fetal Medicine Foundation (FMF) developed a multimodal risk assessment tool for preeclampsia prediction at 11–14 weeks gestation. The development of this tool was a result of extensive research leading to the identification of independent biophysical and biochemical markers useful in predicting preeclampsia. This algorithm includes maternal characteristics, mean arterial blood pressure, and results of uterine artery Doppler examination and maternal serum angiogenic biomarker assessment (pregnancy-associated plasma protein A and/or PlGF [placental growth factor]) [30]. This screening approach was endorsed by the 2019 recommendations of the International Federation of Gynecology and Obstetrics (FIGO) [31]. The high predictive performance of this algorithm was prospectively validated and has been shown to identify 75% of cases of preterm preeclampsia, with a 10% screen-positive rate [32]. The Aspirin for Evidence-Based Preeclampsia Prevention (ASPRE) trial, a double-blind, placebo-controlled trial, was designed to evaluate the effect of 150 mg of aspirin initiated in the first trimester on the rate of preterm preeclampsia [33]. In this seminal study performed in 13 maternity hospitals across six countries, the multimodal FMF assessment tool was used at 11–14 weeks to screen 26,941 European women (over two-thirds were White), after which 1,776 women with a predicted high risk (above 1% risk of preterm preeclampsia), were randomly allocated to receive LDA or placebo. LDA administration reduced the rate of preterm preeclampsia by 62% compared to placebo (1.6% vs. 4.3%, odds ratio [OR] 0.38, CI 0.20–0.74). As maternal serum biomarkers and Doppler assessment may not be feasible in all settings, an alternative is a two-step screening approach, in which the first stage includes screening by maternal factors and mean arterial pressure, reserving measurement of placental angiogenic biomarkers and uterine artery Doppler assessment for the second stage intended for those deemed at high risk following the first stage assessment [31]. This contingent multimodal screening was shown to identify 71% of cases of preterm preeclampsia, with second-stage screening performed in only 30% of the population [34].

As either screening approach may be associated with difficulty in its implementation, and considering the clear beneficial effect of LDA on preterm preeclampsia rates, its low cost, and favorable safety profile, the utilization of universal LDA prophylaxis has been debated [35]. It was shown that universal LDA prophylaxis may be more cost-effective compared to different screening strategies [36]. While some authors believe universal LDA prophylaxis is reasonable [37], others oppose this approach [38]. Universal LDA prophylaxis in the prevention of preeclampsia has not been tested in RCTs, and whether its benefits outweigh its potential risk in increasing maternal and neonatal bleeding is unknown. In addition, adherence rates are likely to be lower when aspirin is given to the whole population, with a presumed smaller beneficial effect. Therefore, at present, universal LDA prophylaxis remains controversial and is neither indicated in clinical practice nor recommended by any major organization.

### 3.2. What Is the Optimal Dose of LDA in the Prevention of Preeclampsia?

The optimal dose of aspirin in the prevention of preeclampsia differs between guidelines and varies between 50–162 mg per day [39]. Roberge et al. demonstrated in a meta-analysis a dose-response effect between aspirin and preterm preeclampsia prevention [20]. The latter study, along with a subsequent meta-analysis that included the ASPRE trial [20,21], indicated that aspirin administered prior to 16 weeks gestation has a beneficial role in the prevention of preterm preeclampsia, only at a daily dose of at least 100 mg. A more recent meta-analysis suggested that a daily dose of 150–162 mg should be preferred [40]. Based on platelet function analysis in 87 pregnant women, a lack of antiplatelet effect occurred in 29% of those who received 81 mg of aspirin, compared to 5% among those who received 162 mg [41]. Additionally, an increase in aspirin dosage from 100 mg prior to pregnancy to 150 mg during pregnancy was required to achieve similar serum salicylic acid concentrations [42]. These findings led the 2019 FIGO guidelines to recommend the use of 150 mg daily, proposing that two tablets of 81 mg are an acceptable alternative, and if either is not available, the minimum dosage of aspirin should be 100 mg/day [31]. The ACOG and the SMFM guidelines continue to recommend a dosage of 81 mg daily but indicate that some practices may consider a higher dosage [22,23,24,43]. Currently, two RCTs (NCT04070573 and NCT05514847) in the United States are planned or underway to evaluate the safety and efficacy of 162 mg vs. 81 mg of aspirin in the prevention of preeclampsia. Finally, it remains to be determined whether aspirin dosage should be weight-adjusted. In a meta-analysis involving non-pregnant subjects, LDA had no vascular benefits in nearly 50% of those weighing more than 70 kg [44]. In a previous RCT among high-risk pregnant women, LDA administered in a daily dose of 60 mg was associated with significantly lower rates of complete inhibition of thromboxane A2 production among obese women [45]. Furthermore, Boelig et al. demonstrated that increased BMI was negatively associated with salicylic acid concentrations during pregnancy [46]. Among the different professional society recommendations, only the FIGO guidelines take maternal body weight into consideration, proposing that in pregnant women weighing less than 40 kg, a daily dose of 100 mg of aspirin is sufficient [31].

### 3.3. What Is the Optimal Timing of LDA Administration during the Day?

Most practice guidelines do not make a recommendation on the optimal time during the day for LDA administration [39], while both the FIGO and the International Society for the Study of Hypertension in Pregnancy (ISSHP) state that aspirin should be given at bedtime [31,47]. In non-pregnant subjects, aspirin given at bedtime reduces platelet activity, which peaks in the morning [48,49] and also diminishes the activity of various regulators of blood pressure (e.g., plasma renin, catecholamines, urinary cortisol) [50]. In a prior randomized chronotherapy trial among 240 pregnant individuals, higher beneficial effects of LDA (100 mg/day) on blood pressure were observed when given at bedtime [51]. A subsequent chronotherapy RCT (N = 350) demonstrated that the positive effects of LDA (100 mg/day) on adverse pregnancy outcomes and blood pressure were maximized when aspirin was administered at bedtime [52]. Based on these findings, in the ASPRE trial, LDA was also taken at night-time [33].

### 3.4. What Is the Optimal Gestational Age for LDA Initiation and Cessation?

Meta-analyses have demonstrated that LDA has to be given prior to 16 weeks in order to have a beneficial effect on preterm preeclampsia rates [20,21]. There is general agreement between different professional society guidelines that aspirin should optimally be initiated prior to 16 weeks (e.g., ACOG and SMFM 12 weeks, FIGO 11–14 weeks) [22,23,31]. However, as a meta-analysis by Meher et al., including 31 RCTs (N = 32,217), found no difference in preeclampsia rates among women who initiated LDA therapy before or after 16 weeks gestation [53], the ACOG and SMFM still recommend commencing LDA in those eligible up untill 28 weeks gestation if not initiated earlier [22,23].

Whether there is a role for LDA initiation prior to 11 weeks or in the pre-conception period remains controversial. An ancillary study to the EAGeR trial demonstrated that elevated levels of platelet factor 4, a marker of platelet activation, are present already prior to pregnancy and associated with increased risk of placenta-mediated complications, particularly hypertensive disorders of pregnancy [54]. A meta-analysis by Chaemsaithong et al., including eight RCTs (N = 1426), demonstrated a non-significant reduction in preeclampsia rates (RR 0.52, CI 0.23–1.117, *p* = 0.115) when LDA was initiated prior to 11 weeks [55]. However, this meta-analysis was limited by the highly heterogonous group of participants among the studies included. This is reflected by the high variability in the dose of aspirin used (50–100 mg) and time of its initiation (from prior to conception up until 8 weeks gestation), with none of the studies including preeclampsia as its primary outcome [55]. Therefore, currently, as neither the safety nor the effectiveness of LDA initiated before 11 weeks of gestation has been demonstrated, it should not be used outside of research protocols prior to 11 weeks in order to prevent preeclampsia.

The suggested timing for LDA cessation differs among professional society recommendations, with some supporting its use until delivery (e.g., ACOG, NICE) [22,23,56], whereas others support its discontinuation earlier (e.g., FIGO 36 weeks, ISSHP 35 weeks) [31,47]. Until recently, this issue has not been investigated. In 2023, the StopPRE trial, a non-inferiority RCT, in which pregnant women with a high risk of preeclampsia, based on the first-trimester multimodal algorithm of the FMF, were treated with daily aspirin at a dose of 150 mg [57]. Those with a sFLT-1/PlGF ratio <38 at 24–28 weeks were randomly assigned (N = 968) to either continue or discontinue LDA treatment at this time point. Aspirin discontinuation at 24–28 weeks was non-inferior to aspirin continuation for preventing preterm preeclampsia and was associated with lower rates of bleeding complications (RR 0.63, CI 0.43–0.93) and a composite of adverse pregnancy outcomes (RR 0.73, CI 0.54–0.98) [57]. A post hoc analysis of the StopPRE trial showed that assessment of uterine artery Doppler at 24–28 weeks and discontinuation of LDA therapy in those with uterine artery pulsatility index <90th percentile can be used instead of sFLT-1/PlGF ratio measurement [58]. Additional studies are needed to establish the effect of earlier discontinuation of LDA treatment and identify subgroups in which such a strategy may be beneficial.

### 3.5. Effects of Aspirin on Other Adverse Outcomes

Given the common pathophysiology, it was suggested that LDA may have a role in preventing other placenta-mediated complications. Previous meta-analyses have suggested lower rates of fetal growth restriction, perinatal death, and placental abruption with LDA therapy [18,20,59]. Other than the ACOG and the SMFM guidelines [22,23], other society guidelines including FIGO [31], consider LDA as a reasonable intervention to prevent fetal growth restriction [60]. In recent years, the possible role of LDA in spontaneous preterm birth prevention has also been reported [61,62,63].

As preeclampsia is associated with adverse long-term maternal cardiovascular outcomes [5,6,7], it is intriguing whether LDA administration during pregnancy will lead to lower rates of cardiovascular events later in life. The most common theory nowadays postulates that preeclampsia is caused by suboptimal cardiovascular adaptation that manifests throughout gestation, with pregnancy as a physiologic stress test [64]. If that is the case, it is unlikely that LDA administration only for the duration of pregnancy will modify the occurrence of cardiovascular disease in the future. On the other hand, if preeclampsia itself leads to cardiovascular damage, then it is plausible that LDA intake throughout gestation may have a role in improving long-term cardiovascular outcomes. Interestingly, in a mouse model of hypertension, placental extracellular vesicles obtained from normotensive term pregnancies reduced long-term hypertensive and cardiovascular disease [65]. Furthermore, improved hypertension control in the early postpartum period among mothers with gestational hypertensive disorders had positive effects on blood pressure measurements and cardiac remodeling up to 9 months following delivery [66,67]. It remains to be further investigated whether LDA administration during pregnancy can have a protective role against long-term maternal cardiovascular disease and whether the FMF screening tool can aid in identifying those at risk for future cardiovascular disease.

### 3.6. What Is the Role of Aspirin in the Prevention of Preeclampsia in Patients with Sickle Cell Disease, Thalassemia, and Myeloproliferative Neoplasms?

Sickle cell disease (SCD) is the most common inherited hemoglobinopathy worldwide. Placenta-mediated complications are more likely to occur in patients with SCD [68,69,70]. A previous meta-analysis found that pregnant patients with SCD have a 2-fold increased risk of preeclampsia, a 3-fold increased risk of fetal growth restriction, and almost a 4-fold increased risk of stillbirth [70]. There is no specific evidence that LDA therapy decreases the risk of preeclampsia in patients with hemoglobinopathies, including SCD. However, in light of the high risk of preeclampsia in this condition, the ACOG [71], the Royal College of Obstetricians and Gynaecologists [72], and other experts [73,74], support the administration of LDA in pregnant patients with SCD starting from 12 weeks gestation. Similarly, in patients with thalassemia major, a more than 2-fold increase in the rate of preeclampsia was reported, and therefore, LDA therapy has also been suggested in this setting [75]. The PIPSICKLE trial is an ongoing RCT (NCT05253781) aimed to determine whether LDA (100 mg/day) versus placebo initiated at 12 weeks can reduce the rates of preeclampsia and fetal growth restriction among patients with SCD.

Myeloproliferative neoplasms (MPN) are often encountered in childbearing age, most commonly essential thrombocythemia and polycythemia vera [76]. The risk of preeclampsia and fetal growth restriction is increased among patients with MPN [77,78]. A meta-analysis of cohort studies among pregnant patients with MPN (N = 1210) showed that LDA therapy is associated with an improved live birth rate (OR 8.6, 95% CI 4.0–18.1) [79]. An additional report, including 121 pregnancies in 52 patients with essential thrombocythemia, demonstrated lower rates of adverse pregnancy outcomes, including preeclampsia, in association with aspirin use (OR 0.29, 95% CI 0.12–0.66) [80]. LDA dose in these studies was highly variable, ranging between 50–160 mg/day [79,80]. Expert groups universally support the administration of LDA during pregnancy in MPN patients [77,78,81,82]. Prior to administration of aspirin in pregnant patients with MPN, it is crucial to measure von Willebrand factor activity level, in order to exclude the presence of acquired von Willebrand disease [83].

In pregnancies among patients with hemoglobinopathies and MPN, the optimal dose and schedule of aspirin therapy during pregnancy require further investigation. Given the supportive data aforementioned in pregnant patients without these conditions, the use of LDA therapy prior to 16 weeks, in a daily dose of at least 100 mg, seems prudent. Currently, guidelines espoused by the different hematological societies do not provide recommendations regarding aspirin use in preeclampsia prevention either for the general population or for those with specific hematologic conditions.

### 3.7. What Is the Optimal Treatment Strategy for Patients Who Experienced Preeclampsia Despite Aspirin Prophylaxis?

An unmet clinical need is how to address patients who experienced preeclampsia despite LDA prophylaxis. In the literature there is no clear definition of what constitutes *aspirin failure* in the prevention of preeclampsia. As aspirin prophylaxis was shown to be effective in preventing preterm preeclampsia, those who suffered from term preeclampsia under aspirin therapy should not be considered non-responders. Moreover, it is critical to ensure the adequacy of aspirin dosage, timing of use, and adherence to treatment. The adjunctive role of low-molecular-weight heparin in those who experienced preterm preeclampsia despite proper utilization of LDA remains controversial and should be considered on a case-by-case basis in a shared decision-making process [84].

## 4. Conclusions

LDA therapy has an important role in the prevention of preterm preeclampsia. Identification of those at risk for preeclampsia should occur early during the first trimester, while the optimal screening approach differs among professional society guidelines. LDA should be administered prior to 16 weeks, with a daily dosage of at least 100 mg given at bedtime in order to maximize its beneficial effects. Future studies are warranted to better delineate the optimal dose of LDA, including in different ethnic or racial groups, the role of its administration prior to 11 weeks, its use in patients with different medical disorders, and the effects of LDA therapy during pregnancy on other adverse pregnancy and long-term outcomes.
jcm-13-04427-t001_Table 1Table 1Key issues related to aspirin therapy in the prevention of preeclampsia.QuestionsSummary of Scientific EvidenceWho should be treated with aspirin in order to prevent preeclampsia? Screening to identify those at high risk for preeclampsia should occur in the first trimesterScreening strategies include:Maternal-characteristics-based screening—simple and identifies only 41% of cases of preterm preeclampsia, with a 64% screen-positive rateMultimodal screening, including maternal characteristics, mean arterial blood pressure, uterine artery Doppler, and maternal serum angiogenic biomarkers—may not be possible in all settings, identifies 75% of cases of preterm preeclampsia, with a 10% screen-positive rateUniversal aspirin prophylaxis is controversial and not indicated in clinical practiceWhat is the optimal dose of aspirin for the prevention of preeclampsia?Aspirin was shown in different meta-analyses to have a beneficial effect on preeclampsia rates only at daily doses of at least 100 mgIt is possible that a daily dose of 150–162 mg may maximize the benefit of aspirinThe role of weight-adjusted dosing remains controversialWhat is the optimal timing of aspirin administration during the day? Prior chronotherapy RCTs have demonstrated that aspirin’s positive effects on adverse pregnancy outcomes and blood pressure were maximized when it was given at bedtimeThe guidelines by FIGO and ISSHP support aspirin administration at bedtimeWhat is the optimal gestational age for aspirin initiation and cessation? Aspirin initiation:Aspirin should optimally be given prior to 16 weeks gestationAs a modest beneficial effect may be present, even if started after 16 weeks gestation, aspirin can be commenced up to 28 weeks if not initiated earlierThe role of aspirin initiation at the pre-conception stage or prior to 11 weeks is controversialAspirin cessation:Guidelines differ in their recommendations for the timing of aspirin discontinuation- some support its use until delivery, others support its discontinuation at 35–36 weeksA recent RCT demonstrated that earlier discontinuation of aspirin at 24–28 weeks may be possible based on maternal serum placental biomarkers and/or uterine artery Doppler assessment. This strategy requires further confirmation and currently is not endorsed by society guidelinesWhat is the effect of aspirin on other adverse outcomesAspirin was shown to have a beneficial effect on other placenta-mediated complications, including fetal growth restriction, perinatal death and placental abruptionAspirin is considered a reasonable intervention to prevent fetal growth restrictionDespite accumulating supportive data, the use of aspirin in order to prevent spontaneous preterm birth is currently not recommended by society guidelinesThe effect of aspirin administered during pregnancy on the occurrence of maternal cardiovascular disease remains unknownWhat is the role of aspirin in the prevention of preeclampsia in patients with sickle cell disease, thalassemia, and myeloproliferative neoplasms?The risk of placenta-mediated complications is increased in patients with hemoglobinopathies and myeloproliferative neoplasmsDespite the lack of specific evidence among patients with SCD and thalassemia major, aspirin therapy is recommended in these conditionsIn patients with myeloproliferative neoplasms, aspirin therapy is recommended as it was shown to improve the rates of live birth and reduce the risk of adverse pregnancy outcomes, including preeclampsiaThe optimal dosing and schedule of aspirin therapy during pregnancy in these hematologic conditions remains to be determinedWhat is the optimal treatment strategy for patients who experienced preeclampsia despite aspirin prophylaxis?There is no clear definition for aspirin failureExperiencing term preeclampsia under aspirin therapy should not be considered a treatment failureIt is critical to ensure the adequacy of aspirin dosage, timing of use, and adherence to treatmentIn those who experienced preterm preeclampsia, despite proper aspirin utilization, adjunctive LMWH should be considered on a case-by-case basisAbbreviations—FIGO—International Federation of Gynecology and Obstetrics; ISSHP—International Society for the Study of Hypertension in Pregnancy; LMWH—low molecular weight heparin; RCT—randomized controlled trial.

## Data Availability

No new data were created or analyzed in this study.

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
