# Peer review of "Controversies and Clarifications Regarding the Role of Aspirin in Preeclampsia Prevention: A Focused Review"

_jcm, 2024, doi:10.3390/jcm13154427_

Round 1
Reviewer 1 Report
Comments and Suggestions for Authors
Dear Author,
This is a very nice mini-review of the use of aspirin in pregnancy to prevent preeclampsia. In my opinion is nicely centered around the FMF approach - the combined PE test at the 11-13 weeks.
There are some things that, in my opinion, would require further elaboration:
Please number your subtitles
In the first chapter - History of aspirin - you briefly cover the mechanism of action of aspirin; in my opinion this should be discussed more as it is of great interest for clinicians: both the general mechanisms of action in cardiovascular prevention but, in particular, in preeclampsia should be detailed more. Please elaborate more on potential mechanisms.
Again - in the second chapter Aspirin in the prevention of preeclampsia - please elaborate more. Maybe the significant RCTs including the ASPRE trial should be explained here: the screening population around 23,000 European women, etc
In the FMF approach an important step is the correct measurement of the mean arterial pressure. I suggest you detail a bit for the general clinicians to understand how and why we use these markers - MAP, PLGF, utPI.
I would suggest including a subtitle on the effect of aspirin in other medical disorders:
- as you know from the secondary analysis of ASPRE, aspirin seems not to be working for chronic hypertension. What is the evidence than that the ACOG uses to reccommend aspirin in CH. What about lupus, diabetes, other auto-immune? IVF, maternal age over 40? Are the doses the same. Some patients with chronic conditions or after IVF receive aspirin 75-100 mg from very early on in the first trimester. Is there evidence for this working? Is there evidence that they should be switched to 150 mg after positive at the FMF screening?
Also, is the aspirin effect the same in different types o f populations? Caucasian vs Afro-carabean, south or east Asian?
I very much liked the point on long term implications for preventing CV disease. The FMF screening in the first trimester may in fact select a population that is high risk for CV disease in later life and they should be targeted and followed accordingly.
Author Response
Reviewer 1:
We thank the Reviewer for the kind words. We believe that the comments contributed significantly to improving our manuscript. We hope that you will find the revised manuscript suitable for publication. Our responses to the specific comments are presented below.
Comment 1:
Please number your subtitles
Text was revised accordingly.
Comment 2:
In the first chapter - History of aspirin - you briefly cover the mechanism of action of aspirin; in my opinion this should be discussed more as it is of great interest for clinicians: both the general mechanisms of action in cardiovascular prevention but, in particular, in preeclampsia should be detailed more. Please elaborate more on potential mechanisms. We agree with Following, the reviewer’s comment we revised this section to highlight the mechanism through which aspirin exerts its antithrombotic mechanism. We also highlight in this section that aspirin positive effect on preeclampsia rates may relate to its antiplatelet and antiinflammatory effects, as well as its direct positive effects on placentation supported by in vitro data showing beneficial effects on trophoblast function, reduction of apoptosis and improvement in placental angiogenic factors.
Comment 3:
Again - in the second chapter Aspirin in the prevention of preeclampsia - please elaborate more. Maybe the significant RCTs including the ASPRE trial should be explained here: the screening population around 23,000 European women, etc
Following the Reviewer’s comment the manuscript was revised accordingly and the ASPRE trial is described in more detail.
Comment 4:
In the FMF approach an important step is the correct measurement of the mean arterial pressure. I suggest you detail a bit for the general clinicians to understand how and why we use these markers - MAP, PLGF, utPI.
Following the Reviewe’s comment, we elaborate on this topic highlighting the extensive research that led to the development of this algorithm.
Comment 5:
I would suggest including a subtitle on the effect of aspirin in other medical disorders:
- as you know from the secondary analysis of ASPRE, aspirin seems not to be working for chronic hypertension. What is the evidence than that the ACOG uses to reccommend aspirin in CH. What about lupus, diabetes, other auto-immune? IVF, maternal age over 40? Are the doses the same. Some patients with chronic conditions or after IVF receive aspirin 75-100 mg from very early on in the first trimester. Is there evidence for this working? Is there evidence that they should be switched to 150 mg after positive at the FMF screening?
We thank the Reviewer for this important comment. We revised the manuscript highlighting the problematic use of aspirin solely based on the presence of different medical conditions. In particularly, we elaborate on pre-gestational diabetes and chornic hypertension, each of which represent a high-risk factor according to the ACOG and SMFM criteria, describing the lack of benefit of aspirin in these conditions according to the several publications.
In addition, the revised version includes a whole section addressing the role of aspirin in certain hematologic disorders.
Comment 6:
Also, is the aspirin effect the same in different types o f populations? Caucasian vs Afro-carabean, south or east Asian?
We agree with the Reviewer. We highlight in the Revised version that further studies are needed in order to better delineate the optimal role and dose of LDA in different ethnic or racial groups.
Comment 7:
I very much liked the point on long term implications for preventing CV disease. The FMF screening in the first trimester may in fact select a population that is high risk for CV disease in later life and they should be targeted and followed accordingly.
We agree with the Reviewer that it will be fascinating to witness whether the FMF screening tool will aid in identifing those at risk for cardiovascular disease in the future. This was added to the revised version.
Reviewer 2 Report
Comments and Suggestions for Authors
In this review, the authors summarized the role of aspirin in preeclampsia prevention. The authors summarized the history of aspirin, its applicability to pregnant women, the optimal dose and timing of taking aspirin, adverse outcomes, and best treatment strategy. The review is well organized and related studies are adequate. Below are a number of issues that the authors shall address or revise:
1. It is better to summarize all these contents within one flowchart, which can give a direct suggestion for aspirin application in preeclampsia prevention.
2. The authors can summarize aspirin studies mentioned in this review in one table. Some key factors should be included, such as the ages of pregnant women, cohort numbers, blood pressure, and effects of aspirin.
3. The authors should give some perspectives on aspirin or other treatments in preeclampsia prevention.
Author Response
Reviewer 2:
We appreciate the Reviewer's comments and editing suggestions which have contributed considerably to the clarity of our manuscript. Our responses are presented below.
Comment 1:
It is better to summarize all these contents within one flowchart, which can give a direct suggestion for aspirin application in preeclampsia prevention.
We thank you for this important suggestion. We include as a supplementary material, a Table which summarizes in a concise manner the different topics highlighted in our paper giving a direct guidance for aspirin application.
Comment 2:
The authors can summarize aspirin studies mentioned in this review in one table. Some key factors should be included, such as the ages of pregnant women, cohort numbers, blood pressure, and effects of aspirin.
We thank the Reviewer for this comment. Our submitted manuscript is a focused and limited review. As a result, we referenced the previous meta-analysis and systematic reviews on this topic, rather than duplicating it. Details of the different studies included as well as their characteristics are included in this references.
Comment 3:
The authors should give some perspectives on aspirin or other treatments in preeclampsia
prevention.
We highlight in this manuscript the history of aspirin, rationale for its use in preeclampsia prevention followed by the clinical data to support aspirin role in this regard. The manuscript also includes the potential role of aspirin in improving long-term maternal outcomes. As the current review is focused on low-dose aspirin, we did not include the scientific evidence to support potential alternative treatments (e.g. aspirin, statins, metformin, PPI, L-arginine, immunomodulatory agents etc.) as this is beyond the scope of our submission. If the Editor and Reviewer would consider changing the focus of the paper, we would be pleased to consider addressing it.